# Variational Techniques for a One-Dimensional Energy Balance Model

Gianmarco Del Sarto[1,2], Jochen Bröcker[3], Franco Flandoli[1], and Tobias Kuna[4]

[1]Scuola Normale Superiore, Pisa, Italy
[2]University School for Advanced Studies IUSS Pavia, Pavia, Italy
[3]University of Reading, Reading, United Kingdom
[4]Università degli Studi dell'Aquila, L'Aquila, Italy

**Correspondence:** Gianmarco Del Sarto (gianmarco.delsarto@sns.it)

**Abstract.** A one-dimensional climate energy balance model (1D-EBM) is a simplified climate model for the zonally averaged global temperature profile, based on the Earth's energy budget. We examine a class of 1D-EBMs which emerges as the parabolic equation corresponding to the Euler–Lagrange equations of an associated variational problem, covering spatially inhomogeneous models such as with latitude-dependent albedo. Sufficient conditions are provided for the existence of at least three steady-state solutions in the form of two local minima and one saddle, that is, of coexisting "cold", "warm" and unstable "intermediate" climates. We also give an interpretation of minimisers as "typical" or "likely" solutions of time-dependent and stochastic 1D-EBMs.

We then examine connections between the *value function*, which represents the minimum value (across all temperature profiles) of the objective functional, regarded as a function of greenhouse gas concentration, and the global mean temperature (also as a function of greenhouse gas concentration, i.e. the *bifurcation diagram*).

Specifically, the global mean temperature varies continuously as long as there is a unique minimising temperature profile, but coexisting minimisers must have different global mean temperatures. Furthermore, global mean temperature is non-decreasing with respect to greenhouse gas concentration and its jumps must necessarily be upward.

Applicability of our findings to more general spatially heterogeneous reaction–diffusion models is also discussed, as are physical interpretations of our results.

## 1 Introduction

### 1.1 Low dimensional energy balance models

Energy balance models are a fundamental tool used to understand the Earth's climate system and its energy dynamics. It represents the energy budget within the Earth's atmosphere, land, oceans, and ice by quantifying the balance between incoming solar radiation and outgoing solar radiation. Although highly simplified compared to general circulation models, EBMs are appreciated for their interpretability, mathematical tractability, and ability to capture the essential dynamics of the Earth's system (Budyko, 1969; Sellers, 1969; North, 1975; Ghil, 1976; Díaz, 1997; Cannarsa et al., 2022). Two important feedback

mechanisms are typically present in such models: the ice-albedo feedback and the Stefan-Boltzmann law. The positive ice-albedo feedback occurs when the melting of ice and snow reduces the surface reflectivity (albedo), causing the planet to absorb more solar radiation. According to the Stefan-Boltzmann law, a warmer body emits more radiation, thereby providing a negative feedback which stabilises the planet's temperature. Depending on the precise configuration, these mechanisms may endow EBMs with bistability, suggesting the existence of two stable climates commonly referred to as the snowball climate and the warm climate. The snowball climate, supported by paleoclimatic evidence from the Cryogenian period around 650 million years ago, is characterized by the absence of vegetation and ice caps extending over the entire planet's surface. In contrast, the warm climate exhibits relatively low albedo, ice caps limited to the polar regions, and the presence of oceans and vegetation. Additionally, EBMs typically allow for a third possible climate, albeit unstable. Transitions between stable climates in an EBM, as well as in general multistable models, can occur in various ways. But two important mechanisms are the following. The first consists of changes in factors influencing the climate system, such as variations in greenhouse gas (GHG) concentrations like carbon dioxide ($CO_2$), altering the balance of incoming and outgoing radiation and amplifying the greenhouse effect. Mathematically, this mechanism can be described by assuming that the model depends on one additional parameter, and changes in the parameter lead the model to undergo a bifurcation (Ashwin et al., 2012); the second consists in noise-induced transitions resulting from unresolved processes in climate models or the representation of short-timescale weather as stochastic forcing acting on slow variables, as observed in stochastic reduced models (Imkeller, 2001; Lucarini et al., 2022). These two types of transitions correspond to mechanisms recognized to induce *climate tipping*, that is rapid non-linear changes in the climate system with potentially irreversible and catastrophic consequences (Lenton et al., 2008; Scheffer et al., 2009; Lenton et al., 2012; Lucarini and Bódai, 2019; Ghil and Lucarini, 2020).

A zero-dimensional (0D) EBM is the simplest version of EBM describing the evolution in time for the annual averaged global mean temperature $T$, without any space dependence (Berger, 1981; North, 1990; North and Kim, 2017; Ghil and Lucarini, 2020). This model is given by an ordinary differential equation (ODE) of the form:

$$C_T \frac{dT}{dt} = \overline{Q}_0 \beta(T) + q - \sigma_0 \varepsilon_0 T^4, \quad t > 0,$$

$$T_{|t=0} = T_0. \tag{1}$$

In this equation, $C_T > 0$ represents the heat capacity, $\overline{Q}_0 > 0$ is the globally averaged solar radiation and the co-albedo $\beta$ is modelled by a continuous function (overbars typically denote globally averaged quantities). Further, $q > 0$ is a positive parameter modelling the effect of the $CO_2$ on the energy budget (Bastiaansen et al., 2022). The term $\overline{R}_e(T) = \sigma_0 \varepsilon_0 T^4$ on the right-hand side of Eq. (1) accounts for the outgoing solar radiation, following the Stefan-Boltzmann law (where $\sigma_0$ denotes the Stefan-Boltzmann constant and $\varepsilon_0$ is the globally averaged emissivity). The fixed points of the model are the solutions of the equation:

$$\frac{dT}{dt} = 0,$$

corresponding to points in Figure 1 where the absorbed radiation $\overline{R}_a(T) = \overline{Q}_0 \beta(T) + q$ and the emitted radiation $\overline{R}_e(T)$ intersect. Figure 1a furthermore illustrates that this model is generally characterized by bistability, with two stable fixed points $T_S$ and $T_W$. These points correspond to the snowball and warm climate states mentioned earlier and are separated by an

unstable intermediate fixed point $T_M$. Furthermore, as highlighted by Figure 1b, the stable points correspond to minimum points of a primitive function $\overline{F}$ for the negative radiation budget $\overline{R}$. In other words, $\overline{F}$ is any regular function such that:

$$\overline{F}'(T) = \overline{R_e}(T) - \overline{R_a}(T) = -\overline{R}(T).$$

To better capture the variability of global mean surface temperature, it has been proposed to add a stochastic forcing, such as white noise, to the radiation balance. This is interpreted as the effect of the fast components of the climate system, i.e. the weather, over slow components (Hasselmann, 1976; North and Cahalan, 1981; Imkeller, 2001; Díaz et al., 2009). For this reason, we are interested in considering the stochastic differential equation (SDE) given by:

$$dT = \overline{R}(T)dt + \varepsilon dW_t, \tag{2}$$

where $\varepsilon > 0$ is the noise intensity and $(W_t)_{t \geq 0}$ is a Brownian motion (Baldi, 2017). This SDE is of gradient type and possesses a unique Gibbs invariant measure $\overline{\nu}$ (Lelièvre and Stoltz, 2016). An invariant measure is a probability distribution $\overline{\nu}$ in the state space of Eq. (2) (i.e. the real numbers in this case) with the property that if a solution $T$ is distributed according to $\overline{\nu}$ at some time $t$ then it remanis so for all later times. It is given by:

$$\overline{\nu}(dT) = \frac{1}{Z} \exp\left(-\frac{2}{\varepsilon^2} \overline{F}(T)\right) dT, \tag{3}$$

where $Z$ is a normalization constant and $dT$ denotes the standard volume element on $\mathbb{R}$ (we note the technical detail that to give meaning to Eq. (2) and Eq. (3), the radiation budget $\overline{R}$ should be extended to negative values for the Kelvin temperature $T$ in a way such that $\overline{F} \to +\infty$ as $T \to -\infty$). The key observation from the explicit formula (3) is that $\overline{\nu}$ is concentrated around the minimum points of the function $\overline{F}$. Indeed, if $T_0$ is a strict minimum point and $T_1 \neq T_0$ is a point close to $T_0$ s.t.

$\overline{F}(T_1) > \overline{F}(T_0)$, then the mass given by the measure $\nu$ in a small neighbourhood of $T_1$ is exponentially lower than the mass around $T_0$; more specifically, the ratio between the two masses is given by $\exp\left(-\frac{2}{\varepsilon^2}\left(\overline{F}(T_1) - \overline{F}(T_0)\right)\right)$.

A one-dimensional (1D) EBM is given by a parabolic partial differential equation where the space variable is one-dimensional (Budyko, 1969; Sellers, 1969; North and Kim, 2017). Denoting the temperature averaged in the zonal direction by $u = u(t, x)$, it extends the 0D-EBM by introducing the sine of the latitude $x = \sin(\phi)$, where $\phi \in \left[-\frac{\pi}{2}, \frac{\pi}{2}\right]$ denotes the latitude and $t \geq 0$

represents time. We assume that the non-linear radiation balance of the planet, denoted by $R(x, u; q)$, depends on the sine of the latitude and on an additive parameter $q$. This parameter models the effect of carbon dioxide concentration on the radiation budget (Bastiaansen et al., 2022). Atmospheric and ocean transport of heat between latitudes is modelled in a very simplified way by a diffusion term. Assuming spatially homogeneous diffusion in this introductory section and thus ignoring the dependence of $\kappa$ on latitude and temperature, we obtain a non-degenerate reaction-diffusion equation:

$$\partial_t u = \kappa \Delta u + R(x, u; q), \quad t > 0, \ x \in (-1, 1),$$

$$u_x(t, -1) = u_x(t, 1) = 0, \quad t \geq 0$$

$$u(0, x) = \tilde{u}(x), \quad x \in [-1, 1] \tag{4}$$

where $\Delta = \partial_{xx}$ denotes the Laplace operator in dimension one, the Neumann boundary conditions impose no-heat flux at the poles and $\tilde{u}$ is an initial condition. The steady-state solutions of this model, representing the asymptotic solutions for the

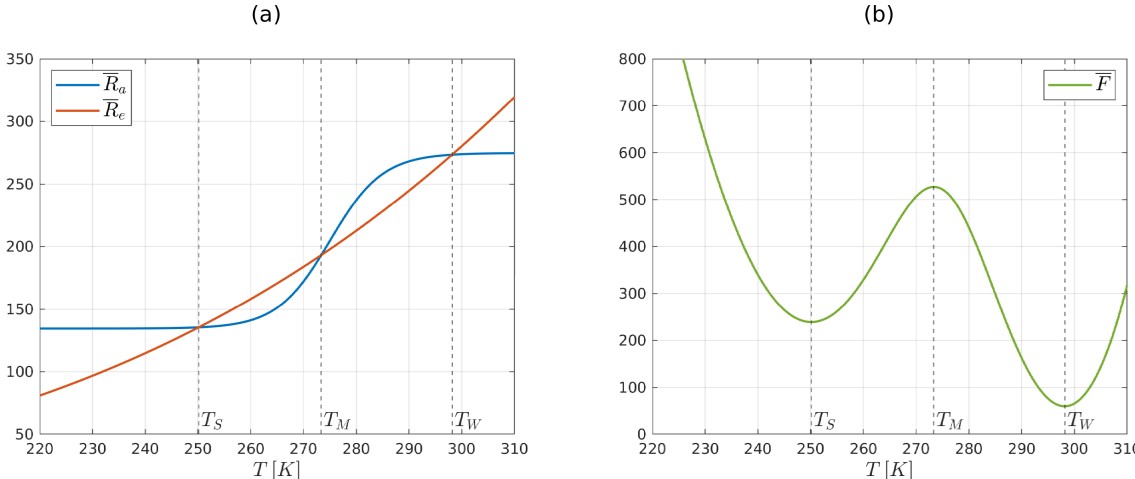

**Figure 1.** (a) Absorbed radiation $\overline{R}_a$ and emitted radiation $\overline{R}_e$ for a 0D-EBM. The graphs intersect in the three fixed points of the model $T_S < T_M < T_W$; $T_S$ and $T_W$ are stable, $T_M$ is unstable. (b) Double-well potential $\overline{F}$ associated to 0D-EBM. The function $\overline{F}$ satisfies $\overline{F}' = \overline{R}_e - \overline{R}_a$. The minimum points $T_S$ and $T_W$ of $\overline{F}$ correspond to stable fixed points.

time-evolving dynamics, correspond to the non-negative solutions $u = u(x)$ of the elliptic problem:

$$0 = \kappa u'' + R(x, u; q), \quad x \in (-1, 1),$$

$$u'(-1) = u'(1) = 0, \tag{5}$$

where $u = u(x)$ depends only on the space variable. This elliptic problem forms a necessary condition for $u = u(x)$ to be our extremal (in particular a local minimiser) for the potential functional

$$F_q(u) = \int\limits_{-1}^{1} \mathcal{R}(x, u; q) dx + \frac{\kappa}{2} ||u'||_2^2 \tag{6}$$

where $\partial_u \mathcal{R}(x, u; q) = -R(x, u; q)$ and $||u'||_2^2 = \int_{-1}^1 (u')^2 dx$ is the square of the norm of $u'$ in $L^2(-1, 1)$. The calculus of variations is a widely employed technique for studying the existence of a solution to the previous problem (North, 1975;

North et al., 1979, 1981; Brezis, 2011). However, proving the existence of a local (but not global) minimum point is generally challenging, and this technique focuses on studying the existence of the global minimum point. The functional $F_q$ in Eq. (6) has another interpretation though which renders it more important than being merely a characterisation of solutions to the elliptic problem. Indeed, consider the stochastic partial differential equation (SPDE) on the Hilbert space $H = L^2(-1, 1)$ given by:

$$du = (\kappa \Delta u + R(x, u; q)) dt + \varepsilon dW_t, \tag{7}$$

obtained by adding a space-time white noise $(W_t)_{t \geq 0}$ modelled by a cylindrical Brownian motion on $H = L^2(-1, 1)$ to Eq. (7). $R$ has a cut-off at negative temperature as in Section 3.1 and $\varepsilon > 0$ is the noise intensity. We refer to Da Prato and Zabczyk

(2014) for more details about SPDEs. It can be shown that this SPDE has a unique invariant Gibbs measure $\nu$ (Da Prato, 2004), given (broadly speaking) by an expression as in Eq. (3), with $F_q$ replacing $\overline{F}$ (see Section 2.1 and 3.1). Therefore, as in the zero-dimensional case, $\nu$ concentrates on minimum points of the functional $F_q$. These minimisers satisfy the elliptic problem (5), which therefore describes temperature profiles around which the solutions of the stochastic problem (7) tend to cluster.

## 1.2 Main results and structure of the paper

This paper focuses on the study of the properties and the interpretation of the steady-state solutions of a 1D-EBM depending on a bifurcation parameter. Motivated by 0D-EBMs, there is a wide consensus in the literature, supported mainly by numerical simulations, regarding the existence of either one or three "interesting" steady-state solutions for 1D-EBMs. Firstly, in Theorem 1 we prove the existence of a steady-state solution for the 1D-EBM by solving the associated variational problem

$$\inf \{F_q(u) \mid u \in \mathbb{X}\},$$

i.e. showing the existence of a global minimum point for the functional $F_q$ over a suitable function space $\mathbb{X}$. Secondly, in Theorem 2 we provide sufficient conditions to have at least three steady-state solutions. These consist of two local minima and one saddle point. The conditions can be summarized as follows:

(i) the viscosity $\kappa$ should be sufficiently large,

(ii) the space-averaged global radiation balance $\mathcal{R}$ of the 1D-EBM should present a double-well potential with sufficiently deep minimum values attained at the two minimum points.

In essence, the conditions require that steady-state solutions of the spatially in-homogeneous 1D-EBM (5) are sufficiently well approximated by steady-state solutions of the spatially *homogeneous* model obtained from spatially averaging the terms in Eq. (5). These assumptions give us the possibility to prove the existence of two minimum points for $F_q$; further, these minimum points are also close to the minimum points of the space-averaged model. Then, the Mountain Pass theorem, a classical result from the calculus of variations, enables us to deduce the existence of a third steady-state solution (Ghil and Childress, 1987; Jabri, 2003). Thirdly, we investigate the uniqueness of the solution of the variational problem in terms of the value function

$$V(q) = \inf \{F_q(u) \mid u \in \mathbb{X}\},$$

which is the minimum value attained by $F_q$ as a function of $q$ which relates to the greenhouse gas concentration. In fact:

(i) in Theorem 3, we show that $V$ is Lipschitz continuous, thus differentiable except for a Lebesgue zero-measure set;

(ii) furthermore, the value function fails to be differentiable if and only if there are two or more co–existing global minimisers for $F_q$. Moreover, $V$ is concave and hence its derivative $V'$ is non-increasing.

(iii) in Corollary 4, we demonstrate how the derivative of $V$ is, up to the sign, the global mean temperature of the global minimum point $u_0$ for $F_q$. This establishes a one-to-one correspondence between the graph of $V$ and the branch of the bifurcation diagram corresponding to $u_0$.

A by–product of our analysis is that the global mean temperature is non-increasing with respect to greenhouse gas concentration $q$. Moreover, it varies continuously with respect to $q$, as long as there is a unique minimising temperature profile. However, for greenhouse gas concentration with coexisting global minimisers, the global mean temperature may exhibit a discontinuous jump as coexisting minimisers cannot all have the same global mean temperature. Furthermore, if a jump occurs, it must necessarily be upward with increasing greenhouse gas concentration.

Our results have a number of interesting physical interpretations. The elliptic 1D-EBM not only describes stationary solutions of the time dependent 1D-EBM but moreover characterises "likely" climates around which the solutions of the stochastic 1D-EBM fluctuate. Global minimisers carry special importance as they are exponentially more likely than just local minimisers. Co-existence of global minimisers is just of special interest as these represent equally likely climate scenarios, and intuitively it seems plausible that rapid transitions between those climates are a dominant feature of the dynamics, although this point

is not investigated further here. Furthermore co-existence of global minimisers imply a discontinuous change of global mean temperature which will jump upwards with increasing greenhouse gas concentration.

  We expect that additional interesting physical conclusions can be drawn through identifying $F_q$ with the (negative) entropy production rate (North and Kim, 2017, Section 7.4.2); this will be subject to future work.

  This paper is organized as follows. In Section 2, we describe the methodology used throughout our work. Firstly, we review

the 1D-EBM proposed in Bastiaansen et al. (2022). This model serves as the reference for our paper and it is characterized by the presence of an additive parameter in the radiation budget, which determines the number of steady-state solutions. Secondly, we recall the properties of the steady-state solutions of the 1D-EBM, that can be obtained from numerical simulations. Finally, we rigorously define the stochastic EBM by introducing space-time white noise. Specifically, we review the invariant measure formula for the resulting reaction-diffusion SPDE. In Section 3, we present our novel findings. In Section 3.1, we discuss the

existence of a solution for the variational problem and outline the properties of the potential functional. Moreover, we explain why the invariant measure of the stochastic EBM concentrates around the global minimum points of the potential functional. Finally, we provide sufficient conditions to demonstrate the existence of at least three steady-state solutions. In Section 3.2, we characterize the uniqueness of the solution to the variational problem in terms of the value function. Additionally, we demonstrate that the value function is Lipschitz, concave and its derivative is non-increasing. In Section 3.3, we illustrate how

knowledge of the value function allows derivation of a portion of the bifurcation diagram and vice versa. In Section 4, we offer a comprehensive summary of our work. In Appendix A, we describe the finite difference method employed to conduct the numerical simulations presented in this study. Furthermore, the Supplementary Material manuscript includes rigorous proofs of our main results.

## 2   Background and methodology

### 2.1   A 1D energy balance model

The fundamental mechanism of 1D-EBMs is that the temperature $u = u(t, x)$, averaged in the zonal direction, evolves in time due to: (i) the diffusion of energy between adjacent regions, (ii) the energy absorbed by the planet, and (iii) the energy emitted

by the planet. The 1D-EBM we consider in this paper is a Seller type EBM where the absorbed radiation depends on an additive parameter (Bastiaansen et al., 2022). We only add a change in the diffusion term in order to get a non-degenerate parabolic PDE. Given an initial condition $\tilde{u}$, the non-linear, parabolic, reaction-diffusion PDE governing the model is given by:

$$C_T \frac{\partial u}{\partial t} = \partial_x \left[ \kappa(x) \partial_x u \right] + R_a(x, u) - R_e(u; q), \quad t > 0, \; x \in (-1, 1)$$

$$\kappa(-1) u_x(t, -1) = \kappa(1) u_x(t, 1) = 0, \quad t \geq 0$$

$$u(0, x) = \tilde{u}(x), \quad x \in [-1, 1], \tag{8}$$

where $R_a$ and $R_e$ represent the radiation absorbed and emitted by the planet per unit area, respectively. $C_T$ is the heat capacity, and the differential term parametrizes the meridional heat transport. The boundary conditions impose no flux at the poles. We now provide further details regarding the parametrization of these terms. The values of the constants of the model can be found in Table 1.

Firstly, the absorbed radiation is assumed to have the form:

$$R_a(x, u) = Q_0(x)(1 - \alpha(u)),$$

where $Q_0$ is the solar radiation per unit area and $\alpha$ is the albedo. The solar radiation is assumed to be

$$Q_0(x) = \hat{Q}_0 \left( c_1 - c_2 x^2 \right), \quad c_i > 0$$

where $\hat{Q}_0$ is the mean solar radiation and $c_i$ are constants. The albedo, which is the proportion of the incident light or radiation that is reflected by a surface, is parametrized by a smooth monotonically decreasing function with a peak derivative in a reference temperature $u_{ref}$ close to the melting point of ice. Specifically

$$\alpha(u) = \alpha_1 + (\alpha_2 - \alpha_1) \left[ \frac{1 + \tanh\left( K(u - u_{ref}) \right)}{2} \right]$$

where $K > 0$ is a rate parameter and $\alpha_1 > \alpha_2$ are respectively the ice-albedo and the water-albedo.

Second, the emitted radiation is modelled using the Stefan-Boltzmann law, in other words assuming that the Earth radiates as a black body. Under this assumption, the energy radiated is proportional to the fourth power of its temperature and it is given by:

$$R_e(u; q) = \varepsilon_0 \sigma_0 u^4 - q,$$

where $\varepsilon_0$ and $\sigma_0$ are respectively the emissivity and Boltzmann's constant. The additive parameter $q$ describes, in a simplified way, the radiative forcing by $CO_2$, i.e. the effect of atmospheric $CO_2$ on the energy budget (Stocker et al., 2014). It is worth explaining: (i) the additive structure of $q$, and (ii) its independence on the spatial variable $x$. About the first point, denote by $C$ the global $CO_2$ concentration in part per million (ppm) and assume, just for explanation purposes, a dependence of the outgoing radiation both on $u$ and $C$. To avoid confusion, we denote by $\hat{R}_e = \hat{R}_e(u, C)$ the outgoing radiation depending on temperature and $CO_2$ concentration. If we linearize $\hat{R}_e$ w.r.t. temperature, we get:

$$\hat{R}_e(u, C) \approx A(C) + B(u - \hat{u}),$$

where $\hat{u}$ is a reference temperature. In Myhre et al. (1998), three radiative transfer models are used in order to get that the dependence of the outgoing radiation which respect to changes in $CO_2$ is given by:

$$A(C) = A_1 - A_2 \cdot \ln\left(\frac{C}{C_0}\right),$$

where $C_0$ is a reference $CO_2$ concentration and $A_1, A_2 > 0$ are explicit constants. In conclusion

$$\hat{R}_e \approx A_1 + B(u - \hat{u}) - q, \quad q = A_2 \cdot \ln\left(\frac{C}{C_0}\right),$$

and thus the radiative forcing of $CO_2$ has an additive structure. About point (ii), we adopt the well-mixed hypothesis for $CO_2$. In other words, we assume that atmospheric $CO_2$ is globally homogeneous, thereby inducing a radiative forcing $q$ independent of latitude (Houghton et al., 2001). This assumption overlooks the spatial pattern of $CO_2$ concentration, which affects many aspects of the climate system, such as the pole-ward heat transport (Huang et al., 2017). It was the state-of-the-art assumption two decades ago, although today it is common to keep in consideration the spatial distribution of radiative forcing (Houghton et al., 2001; Byrne and Goldblatt, 2014; Zhang et al., 2019).

The third component of the model is the term $\partial_x (\kappa(x)u_x)$. It parametrizes the meridional heat transport, that is the phenomenon resulting from the poleward transportation of heat by the Earth-atmosphere system due to the surplus of net radiation heating in the tropics and the deficit in the poleward regions. Usually, the diffusion function $\kappa(x)$ is assumed null at the poles, i.e. with a form such as $\kappa(x) = D(1 - x^2)$, where $D$ is a diffusion constant. This choice is based on the paradigm of mimicking the conduction of heat on a sphere, see North and Kim (2017) for a derivation. On the other hand, it leads to mathematical difficulties in the treatment of the singular PDE arising, in particular in the study of the corresponding variational problem, from which all our results follow. To avoid these difficulties, which at the moment remain an open problem to solve, we add as simplifying assumption that $\kappa$ is non-degenerate and given by:

$$\kappa(x) = D(1 - x^2) + \delta, \quad D, \delta > 0.$$

We choose $\delta = 0.003$, but its value is not important for the results of this work and different choices can be made.

For the parabolic problem (8), the global existence and uniqueness of the solution can be demonstrated, given a regular initial condition (Temam, 1997). Furthermore, if the initial condition is non-negative, the solution remains non-negative for any time $t > 0$. This can be shown proving that $[0, +\infty)$ is an invariant region for Eq. (8), exploiting the fact that there exist $C_1, C_2 > 0$ such that $R(x, u; q) > C_1 > 0$ for all $x \in [-1, 1]$, $u \in [0, C_2]$ (Smoller, 2012).

We recall the formulation of stochastic EBMs using the theory of SPDEs (Da Prato and Zabczyk, 2014). Denote by $\Delta$ the Laplace operator with Neumann boundary conditions. Given an initial condition $\tilde{u} \in H$, we consider the SPDE

$$\partial_t u = \kappa \Delta u + Q_0(x)\beta(u) - R_e(u; q) + \varepsilon dW_t$$

$$u_{|t=0} = \tilde{u} \tag{9}$$

where $\varepsilon > 0$ and $(W_t)_{t \geq 0}$ is a cylindrical Brownian motion on $H$. Under the minor cut-off modifications introduced in Section 3.1, it can be proved that the $H$-valued stochastic process $(u_t)_t$ which solves in mild sense (9) is unique and has continuous

trajectories (Da Prato and Zabczyk, 2014). In addition to this, there exists a unique Gibbs invariant measure

$$\nu(du) = \frac{1}{Z} \exp\left(-\frac{2}{\varepsilon^2} \int\limits_{-1}^{1} \mathcal{R}(x, u; q) dx\right) \mu(du), \tag{10}$$

where $\mathcal{R}$ is as in Eq. (6), $\mu \sim \mathcal{N}(0, -\frac{\varepsilon^2}{2\kappa}\Delta^{-1})$ is a symmetric Gaussian measure on $H$ with covariance $\mathcal{Q} = -\frac{\varepsilon^2}{2\kappa}\Delta^{-1}$ (Da Prato, 2004, 2006). The covariance operator $\mathcal{Q} \colon H \to H$ is the unique linear operator such that $\int_H \langle h_1, \phi \rangle \langle h_2, \phi \rangle \mu(d\phi)$ for each $h_1, h_2 \in H$, where $\langle \cdot, \cdot \rangle$ denotes the scalar product in $H$. Further, it can be shown that $\mathcal{Q}$ is symmetric, positive-definite and its eigenvalues $(\lambda_k)_{k \in \mathbb{Z}}$ satisfy $\sum_{k \in \mathbb{Z}} \lambda_k < \infty$. In the following lines, given a symmetric, positive-definite operator $\mathcal{Q}$ such that $\sum_{k \in \mathbb{Z}} \lambda_k < +\infty$, we are going to explain how to construct an $H$-valued random variable $X$ with law $\mathcal{N}(0, \mathcal{Q})$. Indeed, consider a sequence $(R_k)_{k \in \mathbb{Z}}$ of i.i.d. $\mathbb{R}$-$\mathcal{N}(0, 1)$ random variables defined on a probability space $(\Omega, \mathcal{F}, \mathbb{P})$. We can assume without loss of generality that the eigenvectors $(e_k)_{k \in \mathbb{Z}}$ associated to the eigenvalues $(\lambda_k)_{k \in \mathbb{Z}}$ form an orthonormal basis of $H$. Then, the $H$-valued random variable

$$X = \sum_{k \in \mathbb{Z}} \sqrt{\lambda_k} R_k e_k,$$

is well defined, i.e. the series defining $X$ converges in $L^2(\Omega, \mathcal{F}, \mathbb{P}; H)$, and has law $\mathcal{N}(0, \mathcal{Q})$ (Da Prato, 2006, Proposition 2.18). Further, the convergence also holds $\mathbb{P}$-a.s. in $H$ (Da Prato, 2006, Proposition 2.13).

As mentioned in the introduction, this measure is concentrated on minimum points of the functional $F_q$. A heuristic explanation of this fact can be found in Section 3.1.

The stationary problem associated with the 1D-EBM is given by the elliptic equation for $u = u(x)$:

$$(\kappa(x) u')' + Q_0(x) \beta(u) + q - \varepsilon_0 \sigma_0 u^4 = 0, \quad x \in (-1, 1),$$
$$u'(-1) = u'(1) = 0, \quad u(x) \geq 0. \tag{11}$$

These solutions can be either stable or unstable, depending on the long-term behaviour of their infinitesimal perturbations. As pointed out in Bastiaansen et al. (2022), if the reaction-diffusion equation for $u = u(t, x)$ was space-homogeneous, i.e. of the form:

$$\partial_t u = \kappa \Delta u + R(u), \tag{12}$$

then the stable steady-state solutions would correspond to functions that are constant in space and time, with values given as the roots of

$$R(y) = 0.$$

A rigorous result in this direction has been shown in Gaspar and Guaraco (2018). Indeed, for a fixed double-well symmetric potential, it has been proved that: (i) if $\kappa$ is large enough, the only steady-state solutions of (12) are the constants where the potential is critical, and (ii) the number of unstable steady-state solutions to (12) can be made arbitrary large as $\kappa \to 0$. Introducing a spatial dependence in $R = R(x, u)$ leads to a space-heterogeneous model. Depending on the space heterogeneity, it can exhibit any number of both stable and unstable steady-state solutions (Bastiaansen et al., 2022). The variational approach

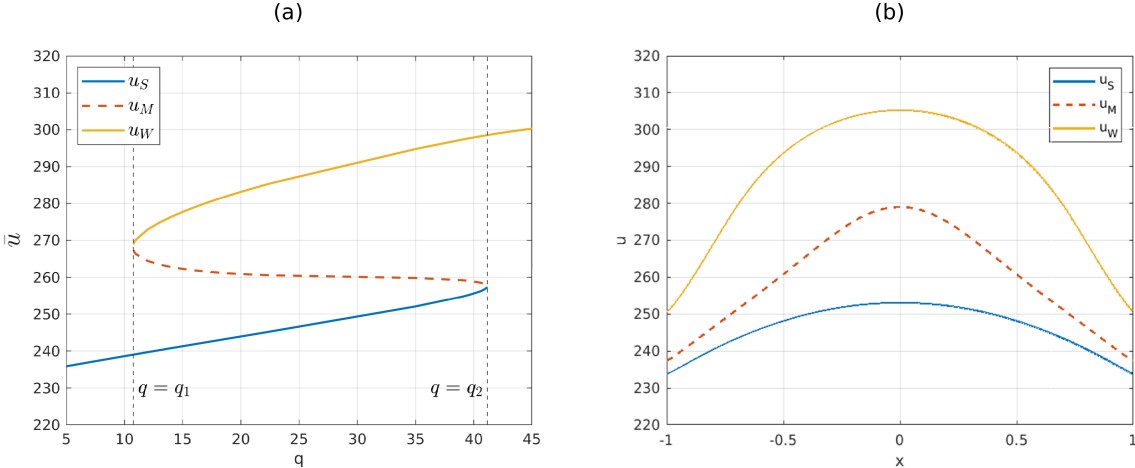

**Figure 2.** (a) Bifurcation diagram of the steady-state solutions in the $(q, \bar{u})$ plane, with $\bar{u} = \int_{-1}^{1} u(x)dx$. Solid lines denote stable solutions $u_S$ and $u_W$, while dashed lines the unstable solution $u_M$. (b) Steady-state solutions of the EBM for $q = 25$. In every point $x$ of the space domain, the three steady-state solutions satisfy $u_S(x) < u_M(x) < u_W(x)$, with maximum temperature attained at the equator and minimum temperature attained at the poles.

to the study of steady-state solutions provides a tool for characterizing the stable ones, which are the local minimum points of
a functional.

    In the following paragraph, we describe the properties of the solutions of (11). As the parameter $q$ changes, numerical simulations for Eq. (11) suggest the existence of either one or three steady-state solutions. That is, there exists $q_1 < q_2$ s.t. Eq. (8) has one steady-state solution if $q < q_1$ or $q > q_2$, and the steady-state solutions are three if $q_1 \leq q \leq q_2$. In the latter case, we denote the solutions by $u_S \leq u_M \leq u_W$, corresponding respectively to the snowball climate, a middle (or intermediate)
climate and the warm climate. As an analogy, we denote by $u_S$ the unique steady-state solution for $q < q_1$ and by $u_W$ the unique one for $q > q_2$. Figure 2a shows the bifurcation diagram of the model in the $(q, \bar{u})$ plane, where $\bar{u} = \int_{-1}^{1} u(x)\, dx$ denotes the average temperature. Figure 2b depicts the three steady-state solutions for $q = 25 \in (q_1, q_2)$. A stability analysis can be conducted to determine the stability of the steady-state solutions. The results show that $u_S$ and $u_W$ are stable, while the middle climate $u_M$ is unstable. Furthermore, it's worth noting that special values $q = q_1, q_2$ correspond to bifurcation points of
saddle-node type, where the unstable solution $u_M$ collides with either $u_W$ (for $q = q_1$) or $u_S$ (for $q = q_2$) and then disappears. These numerical findings regarding the number and stability of the steady-state solutions will be supported and validated using rigorous arguments, as in the next section.

**Table 1.** Parameters and constants appearing in the Seller EBM (8).

| Symbol | Meaning | Value |
|:------:|:-------:|:-----:|
| $D$ | Diffusivity constant | 0.3 |
| $\delta$ | Perturbation constant- meridional heat transport parametrization | 0.003 |
| $\hat{Q}_0$ | Mean solar radiation | $341.3\,\mathrm{W\,m^{-2}}$ |
| $\varepsilon_0$ | Emissivity | 0.61 |
| $\sigma_0$ | Boltzmann's constant | $5.67 \cdot 10^{-8}\mathrm{W\,m^{-2}\,K^{-1}}$ |
| $\alpha_1$ | Ice albedo | 0.7 |
| $\alpha_2$ | Water albedo | 0.289 |
| $K$ | Constant rate - albedo parametrization | 0.1 |
| $u_{ref}$ | Reference temperature - albedo parametrization | 275 K |
| $C_T$ | Heat capacity | $5 \cdot 10^8\,\mathrm{J\,m^{-2}K^{-1}}$ |

## 3 Results

### 3.1 Potential functional and its minimiser

In this section, we: (i) provide an intuitive motivation for why the invariant measure for the stochastic EBM concentrates on minimum points of the functional $F_q$, (ii) prove the existence of global minimum points for $F_q$ using the direct method, (iii) present sufficient conditions on the viscosity $\kappa$ and the space-averaged potential $\bar{\mathcal{R}}(u) = \frac{1}{2}\int_{-1}^{1} \mathcal{R}(x,u)dx$, with $\partial_u \mathcal{R} = -R = R_e - R_a$, to ensure that the 1D-EBM has at least three steady-state solutions.

     Firstly, consider the stochastic EBM (9). Assume that for a negative value of $u$, where the model has no physical meaning, 210   the Stefan-Boltzmann law is extended as:

$$R_e(u) = \begin{cases} \varepsilon_0\sigma_0 u^4, & \text{if } u \geq 0 \\ 0, & \text{if } u < 0. \end{cases}$$

and $\beta$ is smoothly extended to $\tilde{\beta}$ by setting it to zero outside the physically relevant range, as described in the Supplementary Material. Then, Eq. (9) possesses a unique Gibbs invariant probability measure given by:

$$\nu(du) \propto \exp\left( -\frac{2}{\varepsilon^2}\int_{-1}^{1} \varepsilon_0\sigma_0 \frac{(u^5)_+}{5} - Q_0(x)B(u) - qu\,dx \right)\mu(du), \quad \mu \sim \mathcal{N}(0, -\frac{\varepsilon^2}{2\kappa}\Delta^{-1}), \tag{13}$$

where $(u)_+ = \max\{u,0\}$ is the positive part, $\mathcal{N}(0, -\frac{\varepsilon^2}{2}\Delta^{-1})$ denotes a symmetric gaussian measure with covariance operator $\mathcal{Q} = -\frac{\varepsilon^2}{2\kappa}\Delta^{-1}$ over the Hilbert space $H = L^2(-1,1)$ and $Z$ is the normalization constant. See Da Prato (2004) for a rigorous derivation of the invariant measure for a reaction-diffusion model with a polynomial homogeneous reaction term. We move to

explain in what sense $\nu$ is concentrated around minimum points of $F_q$. In fact, for $u \in H$ the gaussian measure $\mu$ is formally given by:

$$\mu(du) = \frac{1}{Z_1} \exp\left(-\frac{1}{2}\langle \mathcal{Q}^{-1}u, u\rangle\right) du,$$

where $\mathcal{Q}^{-1} = -\frac{2\kappa}{\varepsilon^2}\Delta$. Here, $Z_1$ is a normalization constant, $\langle \cdot, \cdot \rangle$ denotes the scalar product in $H$, and $du$ is a formal notation for the Lebesgue measure on $H$. If we perform an integration by parts, we get

$$\mu(du) = \frac{1}{Z_1} \exp\left(\frac{\kappa}{\varepsilon^2}\langle u'', u\rangle\right) du = \frac{1}{Z_1} \exp\left(-\frac{\kappa}{\varepsilon^2}\|u'\|_2^2\right) du.$$

Plugging the previous identity into Eq. (10), we obtain:

$$\nu(du) \propto \exp\left(-\frac{2}{\varepsilon^2}\left(\int_{-1}^{1} \varepsilon_0\sigma_0\frac{(u^5)_+}{5} - Q_0(x)B(u) - qu\,dx + \frac{\kappa}{2}\|u'\|_2^2\right)\right) du$$

$$\propto \exp\left(-\frac{2}{\varepsilon^2}F_q(u)\right) du.$$

From this heuristic formula, we see that points $u$ such that $F_q(u)$ is not a global minimum have exponentially smaller density than the minimum points. Indeed, if $u_1$ is a global minimum point and $u \neq u_1$, then the mass given by $\nu$ in a small neighbourhood around $u$ is exponentially smaller than the mass given to a neighbourhood of the same size around $u_1$; in particular, the ratio between the two masses is given by $\exp\left(-\frac{2}{\varepsilon^2}\left(F_q(u) - F_q(u_1)\right)\right)$. The previous derivation is formal because the Lebesgue measure cannot be defined on an infinite dimensional Hilbert space. For a more rigorous explanation, see Section S2 in the Supplementary Material.

Next, we discuss the properties of the functional $F_q \colon H^{1,2}(-1,1) \cap \{u \geq 0\} \to \mathbb{R}$ given by:

$$F_q(u) = \int_{-1}^{1} \frac{u^5}{5}\varepsilon_0\sigma_0 - Q_0(x)B(u) - qu\,dx + \frac{1}{2}\int_{-1}^{1} \kappa(x)(u'(x))^2\,dx,$$

where $B$ is a primitive of the co-albedo $\beta(u) = 1 - \alpha(u)$ and $H^1 = H^{1,2}(-1,1)$ denotes the Sobolev space of order 1 and exponent 2, i.e. the function space where a function $u$ and its derivative $u'$ (in weak-sense) are both square integrable over $[-1,1]$. See Brezis (2011) for more details about Sobolev spaces. The functional $F_q$, depending on the parameter $q$, is known in the literature as potential functional or Lyapunov function (North et al., 1979; North and Kim, 2017). The study of the functional $F_q$ gives useful information thanks to its links with the invariant measure for the stochastic 1D-EBM, as we have seen, and the stable steady-state solutions for the deterministic 1D-EBM which emerge as necessary conditions for the stationarity of $F_q$. Going deeper with the former point, the first variation of $F_q$ in the point $u$ in direction $h$ is given by:

$$\delta F_q(u,h) = \frac{d}{ds}F_q(u+sh)_{|s=0} = \int_{-1}^{1}\left(u^4\varepsilon_0\sigma_0 - Q_0(x)\beta(u) - q\right)h\,dx + \int_{-1}^{1}\kappa(x)u'(x)h'(x)dx$$

$$= \int_{-1}^{1}\left[u^4\varepsilon_0\sigma_0 - Q_0(x)\beta(u) - q - (\kappa(x)u'(x))'\right]h\,dx$$

where in the last identity we have used the integration by parts. Since $h$ is arbitrary, $u$ is a stationary point for the functional $F_q$ if and only if it is a steady-state solution for the EBM. In particular, local extremum points for $F_q$ correspond to steady-state solutions of the EBM. Any local minimiser of $F_q$ represents a locally attractive solution of the deterministic 1D-EBM. In view of our interpretation of $F_q$ in terms of the invariant measure, however, global minimisers play a special role since if present and unique they are exponentially more likely than any other state (including minimisers that are just local). The following result establishes the existence of a global minimum point for $F_q$.

**Theorem 1.** *If $q > 0$, then there exists a global regular non-negative minimiser for $F_q$. In other words, if we consider the variational problem*

$$\inf \left\{ F_q(u) \mid u \in H^1, \, u \geq 0 \right\}, \tag{14}$$

*then there exists $u_0 \in C^\infty$ s.t. $u_0$ is a solution of the EBM and*

$$F_q(u_0) = \inf \left\{ F_q(u) \mid u \in H^1, \quad u \geq 0 \right\}.$$

*In addition to this, if $q$ belongs to a bounded interval, then $u$ can be bounded uniformly with respect to $q$:*

$$\exists M > 0 \text{ s.t. } u_0(x) \leq M, \quad \forall x \in [-1, 1]. \tag{15}$$

A rigorous proof of the previous result can be found in Section S3 of the Supplementary Material manuscript. The proof relies on standard arguments from the direct method of calculus of variation exploiting the fact that the outgoing radiation in the EBM model prevents the temperature from being too high.

Concerning the existence of two local minimum points, let us describe a sufficient condition. Consider the potential function $\bar{\mathcal{R}} \colon \mathbb{R} \to \mathbb{R}$ coming from the space averaged model

$$\bar{\mathcal{R}}(u) = \frac{1}{2} \int\limits_{-1}^{1} \mathcal{R}(x, u) dx.$$

If the viscosity $\kappa > 0$ is sufficiently large and the function $\bar{\mathcal{R}}$ has a double well shape with sufficiently deep minimum values attained at the minimum points, then we are able to prove the existence of two minimum points for $F_q$. Further, it is possible to prove that the functional $F_q$ satisfies a compactness condition known as Palais-Smale condition. This property and the Mountain Pass theorem give the possibility to deduce the existence of a third steady-state solution. Next, we characterize a situation in which there are three steady-state solutions, two of which are local minimisers (Jabri, 2003). This is summarized in the following result.

**Theorem 2.** *Denote by $B_{H^1}(v, \rho) = \{u \in H^1 \mid ||u - v||_{H^1} < \rho\}$ the open ball in $H^1$ with center $v$ and radius $\rho > 0$. Assume $\bar{\mathcal{R}}$ has two non-negative minimum points $u_1 \neq u_2$, with $F_q(u_1) \geq F_q(u_2)$. Then, there exist $\omega > 0$ and $f, g \in O(\varepsilon^{-1})$ as $\varepsilon \to 0^+$ s.t. if $\bar{\varepsilon} > 0$ satisfies:*

*(i) $\bar{\mathcal{R}}''(u_i) > f(\bar{\varepsilon})$, for $i = 1, 2$,*

*(ii)* $\kappa > g(\bar{\varepsilon})$,

*(iii)* $\bar{\varepsilon} \leq \omega$,

then $\tilde{F}_q$ has two local minimum points $\tilde{u}_1, \tilde{u}_2$ such that:

*(a)* $B_{H^1}(u_1, \bar{\varepsilon}) \cap B_{H^1}(u_2, \bar{\varepsilon}) = \emptyset$,

*(b)* $\tilde{u}_i \in B_{H^1}(u_i, \bar{\varepsilon})$, for $i = 1, 2$,

*(c)* If $\|u - u_1\|_{H^1} = \bar{\varepsilon}$, then $F_q(u) \geq F_q(u_1) + \delta$, with $\delta = \delta(\bar{\varepsilon}) > 0$.

Note how the previous result can be also interpreted as giving sufficient conditions for the convergence of the stable solutions of a space-inhomogeneous EBM to the stable solution of the corresponding space-averaged model, as the diffusion becomes large.

## 3.2 Value Function and uniqueness for the functional minimiser

The key element of this section is the value function, which is given by:

$$V(q) = \inf \left\{ F_q(u) \mid u \in H^1,\, u \geq 0 \right\}.$$

From Section 3.1, we know that the previous infimum is indeed a minimum and so $V(q)$ can be interpreted as the minimum possible value attained by the potential functional over the possible temperature profiles $u$. Since a minimum point for $F_q$ is also a stationary point for the functional, the value function can be evaluated numerically by computing the minimum of the three steady-state solutions $u_S, u_M, u_W$. Following this strategy, Figure 3 shows $q \mapsto F_q(u_*)$, with $u_* \in \{u_S, u_M, u_W\}$. Particularly, there exists a point $q_3$ s.t. $u_S$ it the global minimum point of $F_q$ for $q < q_3$, while $u_W$ is the global minimum point for $q > q_3$. Further, for $q = q_3$ the function $F_q$ has two different global minimum points $u_S, u_W$ and $q = q_3$ correspond a non-differentiability point for $V$. In addition to this, the value function appears to be concave, thus with a decreasing derivative, where it exists. Summarizing, the numerical evaluations of $V(q)$ suggest the following result, a rigorous proof of which is included in the Supplementary Material.

**Theorem 3.** *Assume $q$ belongs to a bounded interval. Then:*

*(i) $V$ is Lipschitz continuous.*

*(ii) $q$ is a non-differentiable point for $V$ if and only if there is more than one minimiser for $F_q$.*

*(iii) $V$ is concave and $V'$ is non-increasing.*

We also see numerically that $u_M$ is actually never a global minimiser for the specific functional $F_q$ considered here, but we do no have rigorous proof of this fact. Let's briefly discuss the proofs of the previous points. The proof of (i) follows from the facts that the sup-norm of the minimiser $u_0$ can be bounded uniformly in $q$ and that, given a family $\{g_i\}_{i \in I}$ of $L_i$-Lipschitz

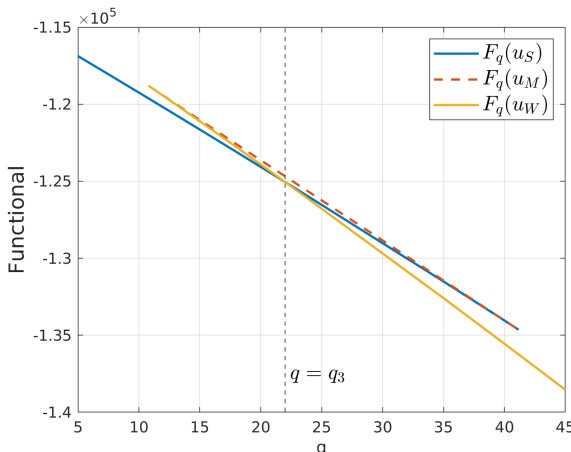

**Figure 3.** Potential functional $F_q$ evaluated in the three steady-state solutions $u_S, u_M, u_W$. For $q < q_3$, $u_S$ is the global minimum point, while $u_W$ is a local minimum point. On the other hand, for $q > q_3$ the vice versa happens. Solid lines correspond to values of the functional attained on stable solutions, dashed lines for values corresponding to unstable ones.

functions $g_i$, then $\inf_{i \in I} g_i$ is Lipschitz if the constants $L_i$ can be bounded uniformly. In our case, given $u \in H^1$ non-negative, we have

$$|F_{\mu_1}(u) - F_{\mu_2}(u)| \leq |\mu_1 - \mu_2| \int_{-1}^{1} |u(x)| \, dx \leq 2M|\mu_1 - \mu_2|$$

where $M > 0$ is the constant appearing in Eq. (15). On the other hand, the proof of point (ii) is less straightforward, although being very similar to the one for the existence of a solution for the variational problem. The proof of point (iii) makes use of the concept of semiconcavity, a generalisation of that of concavity, which is fundamental in optimal control (Cannarsa and Sinestrari, 2004). The main reason for the concavity of $V$ though is that $V$ is an infimum over functions that are affine in $q$. Hence the fact that $q$ is additive is essential for this result. More details can be found in Section S4 and S5 of Supplementary Material.

## 3.3 Value function graph and bifurcation diagram

An additional property of the value function can be observed when comparing the bifurcation diagram (Figure 4a) and the graph of the value function (Figure 3).

**Corollary 4.** *If $V$ is differentiable, then $V'(q) = - \int_{-1}^{1} u_0(x) dx$, where $u_0$ is the only minimiser for $F_q$.*

In other words, the part of the bifurcation diagram that corresponds to the global minimiser, represented by the subgraph $(q, \int_{-1}^{1} u_0(x), dx)$, can be determined based on the knowledge of $V'$, and vice versa. Figure 4 compares Figure 2a and Figure 3, highlighting in magenta the corresponding parts of the two graphs. From the mathematical point of view, the previous result is a consequence of the proof of Theorem 3.

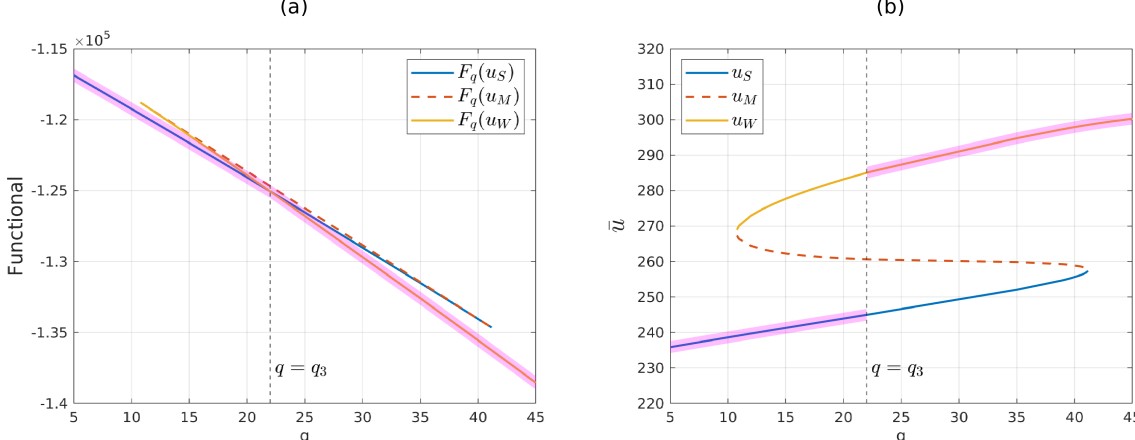

**Figure 4.** Comparison between the value function graph (left) and bifurcation diagram (right) for the 1D-EBM. The magenta-shaded area highlights the parts of the plots which are in one-to-one correspondence. (a) Functional $F_q$ evaluated on steady-state solutions, as in Figure 3. (b) Bifurcation diagram, as in Figure 2a.

It is worth pointing out that by combining Theorem 3 and Corollary 4, a valuable property emerges, i.e., the global mean temperature of the functional minimiser is non-decreasing with respect to $q$. In other words, as the concentration of $CO_2$ rises, the global mean temperature increases. Additionally, through this monotonicity and Froda's theorem (Rudin, 1976, Theorem 4.30), we also establish that the global mean temperature is continuous, except for, at most, a countable number of upward jumps.

In the second part of this section, we demonstrate the applicability of Corollary 4 to other reaction-diffusion equations. We use as an example a spatially heterogeneous Allen-Cahn equation (ACE), already considered in Bastiaansen et al. (2022). For an initial condition $\tilde{u}$, this model is given by:

$$\partial_t u = \frac{1}{100}\Delta u + u(1 - u^2) + q + \frac{1}{2}\cos(\pi x), \quad x \in (-1, 1),\, t > 0,$$
$$u_x(t, -1) = u_x(t, 1) = 0, \quad t \geq 0,$$
$$u_{|t=0} = \tilde{u}. \tag{16}$$

The associated elliptic problem for $u = u(x)$ is

$$0 = \frac{1}{100}u'' + u(1 - u^2) + q + \frac{1}{2}\cos(\pi x), \quad x \in (-1, 1),$$
$$u'(-1) = u'(1) = 0. \tag{17}$$

In this case, the potential functional takes the form

$$J_q(u) = \int_{-1}^{1} \frac{u^4(x)}{4} - \frac{u^2(x)}{2} - u(x)(q + \frac{1}{2}\cos(\pi x))\, dx,$$

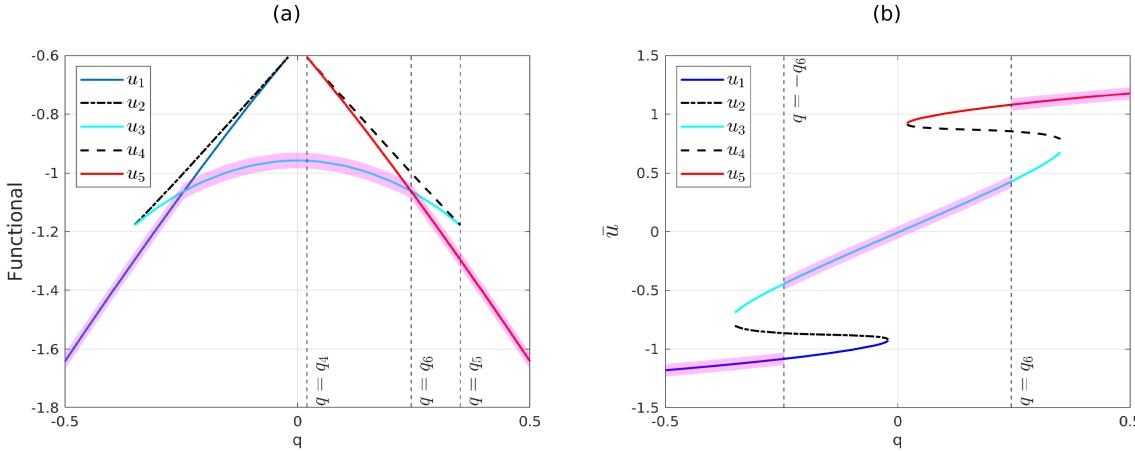

**Figure 5.** Comparison between the value function and the bifurcation diagram for the non-homogeneous ACE. The magenta-shaded area highlights the parts of the plots which are in one-to-one correspondence. (a) Potential functional evaluated on the steady-state solutions: $u_1$ is the global minimum point for $q < -q_6$, $u_3$ is the global minimum point for $-q_6 < q < q_6$, $u_5$ is the global minimum point for $q > q_6$. Note that $q = \pm q_6$ are the non-differentiability point for the value function, corresponding to non-uniqueness of the minimiser. (b) Bifurcation diagram.

and all the properties discussed in Section 3.1 and 3.2 can be extended to this equation. Specifically, Theorems 1, 2 and 3 hold. But in this case, the structure of the bifurcation diagram is more complex, even if symmetric which respect to $q = 0$. Indeed, through numerical experiments, it is possible to deduce the existence of $0 < q_4 < q_5$ such that: (a) for $|q| > q_5$ or $|q| < q_4$, there exists a single steady-state solution, which is stable, (b) for $q_4 < |q| < q_5$, there are three steady-state solutions, two of which are stable while the third is unstable. Further, $q = q_4, q_5$ are bifurcation points of saddle-node type. We denote by $u_1$ the steady-state solution for $q < -q_5$, by $u_2, u_3$ the steady-state solutions appearing at the bifurcation point $q = -q_5$ and existing for $-q_5 < q < -q_4$ in addition to $u_1$ and by $u_4, u_5$ the steady-state solutions appearing at $q = q_4$ and existing for $q_4 < q < q_5$ in addition to $u_3$. Regarding the potential functional $J_q$, in this case there exists $q_6 \in (q_4, q_5)$ such that $u_1$ is the global minimum point for the functional for $q < -q_6$ and $u_3$ is the global minimum point for $-q_6 < q < q_6$, while $u_5$ becomes the global minimum point for $q > q_6$. A picture for the bifurcation diagram just described and the value function is shown in Figure 5. Note that $q = \pm q_6$ are the only values of the parameter $q$ for which the value function is not differentiable and also the only points in which the global minimiser of the variational problem is not unique.

## 4 Conclusions

In this paper, we have considered a one-dimensional energy balance model depending on a bifurcation parameter $q$, describing the effect of $CO_2$ concentration in the atmosphere and affecting the energy absorbed by the planet. Numerical simulations show that this model can exhibit either one or three asymptotic solutions, depending on the values of $q$. We began our analysis by introducing the potential functional $F_q$ associated with the steady-state solutions. The functional $F_q$ has significant implications,

as it is closely linked to both the stability of steady-state solutions of the EBM and the invariant measure for the stochastic EBM obtained by perturbing the model with an additive Gaussian white noise. In particular, the invariant measure of the system concentrates on global minimisers of $F_q$, giving them exponentially larger weight than local minimisers. By analyzing the first variation of $F_q$ and applying standard arguments from the direct method of calculus of variations, we established that $F_q$ possesses a global regular minimiser for all values of the parameter $q$. Furthermore, we provide sufficient conditions to prove the existence of at least three steady-state solutions for the 1D-EBM.

We then introduced the value function $V(q)$, which represents the minimum value attained by the potential functional among all possible temperature profiles. By evaluating $V(q)$ numerically using the steady-state solutions $u_S, u_M, u_W$, we observed that the function exhibits Lipschitz continuity and concavity. Furthermore, non-differentiability points of $V(q)$ coincide with points where multiple global minimisers exist for $F_q$. Lastly, when $V$ is differentiable, its derivative is non-increasing and equal to the negative global mean temperature, i.e. $V'(q) = -\int_{-1}^{1} u_0(x)\,dx$, where $u_0$ is the minimiser for $F_q$. Moreover, as a consequence of the explicit expression for $V'$, the global mean temperature is non-decreasing with respect to $q$ and it is continuous, except for a Lebesgue zero-measure set of upwards jumps. These are the non-differentiability points of $V$, corresponding to the case where two or more global minimisers, hence multiple climate equally probable, exist for the stochastic EBM. These findings, which we are able to prove rigorously, allow us to establish a correspondence between the bifurcation diagram and the graph of the value function. Additionally, we applied our results to a spatially inhomogeneous Allen-Cahn equation, to show how our results still hold for more general space-inhomogeneous reaction-diffusion equations.

The diffusion function $\kappa = \kappa(x)$ that we have examined is non-degenerate at the boundary of the spatial domain. As noted in Section 2.1, this is an assumption to simplify the study of the variation problem. At present, there remains a problem with how to extend our results to the case where $\kappa$ is degenerate at the boundary.

Regarding the impact of our work on current climate change, we have characterized climate as an invariant measure within a stochastic equation that describes temperature. The emission of $CO_2$ is considered a parameter influencing the shape of this invariant measure, particularly in relation to the points around which the measure is concentrated. From our perspective, the climate we are currently witnessing reflects changes in the invariant measure, representing a realization of a random variable with that invariant measure as its distribution. Moreover, we have demonstrated the monotonic relationship between global mean temperature and $CO_2$. Finally, we have outlined simple conditions, adaptable to other multi-stable reaction-diffusion models, to establish the existence of three asymptotic climate states.

Concerning future development of this work, one interesting aspect we are working on is to understand how the invariant measure for the stochastic EBM changes close to bifurcation points. This points in the direction of using statistical indicators to detect the approach of tipping points, which in our model correspond to points of discontinuity of the global mean temperature with respect to the parameter $q$.

*Data availability.* This work does not include any externally supplied code, data, or other material. All material in the text and figures was produced by the authors using standard mathematical and numerical analysis by the authors. The code is available at Zenodo (DOI: 10.5281/zenodo.10469451).

## 360 Appendix A: Numerical methods

In this section, we describe the numerical method adopted to approximate the solutions of the elliptic problem (11) numerically. We used a classical finite difference scheme, which we are going to illustrate (Quarteroni and Valli, 2008; Thomas, 2013). To simplify the notation, let's define $f(x,u) = R_e(x,u) - R_a(u)$ the non-linear reaction term. We consider a uniform mesh for $[-1,1]$ made of $n+1$ points

$$x_0 = -1 < x_1 < \cdots < x_n = 1, \quad x_i = -1 + i\Delta x, \quad i = 0,...,n, \quad \Delta x = \frac{2}{n}.$$

Then, the solution to the problem can be approximated by considering the system

$$\frac{u_{i-1}\kappa_{i-\frac{1}{2}} - u_i(\kappa_{i-\frac{1}{2}} + \kappa_{i+\frac{1}{2}}) + u_{i+1}\kappa_{i+\frac{1}{2}}}{\Delta x^2} + f(x_i, u_i) = 0, \quad i = 0,...,n$$

$$\frac{u_1 - u_{-1}}{2\Delta x} = \frac{u_{n+1} - u_{n-1}}{2\Delta x} = 0$$

where $u_{-1}, u_{n+1}$ are ghost points and $u_i = u(x_i)$, $\kappa_{i\pm\frac{1}{2}} = \kappa(x_{i\pm\frac{1}{2}})$, $x_{i\pm\frac{1}{2}} := x_i \pm \Delta x/2$. The system of equations can be written in vector form as:

$$\frac{1}{\Delta x^2}\begin{bmatrix} -\Delta x/2 & 0 & \Delta x/2 & & & \\ \kappa_{-1/2} & -(\kappa_{-1/2}+\kappa_{1/2}) & \kappa_{1/2} & & & \\ & \ddots & \ddots & \ddots & & \\ & & \kappa_{n-1/2} & -(\kappa_{n-1/2}+\kappa_{n+1/2}) & \kappa_{n+1/2} \\ & & -\Delta x/2 & 0 & \Delta x/2 \end{bmatrix}\boldsymbol{u} + \begin{bmatrix} 0 \\ f_0 \\ \vdots \\ f_n \\ 0 \end{bmatrix} = \boldsymbol{0},$$

with $\boldsymbol{u} = \begin{bmatrix} u_{-1}, \cdots u_{n+1} \end{bmatrix}^T$ and $f_i = f(x_i, u(x_i))$. At this point, multiplying the first equation by $2\frac{\kappa_{1/2}}{\Delta x}$, subtracting the second one and dividing by 2, we get

$$-\frac{\kappa_{-1/2}+\kappa_{1/2}}{2}u_{-1} + \frac{\kappa_{-1/2}+\kappa_{1/2}}{2}u_0 - \frac{f_0}{2} = 0.$$

In a symmetric way, multiplying the last equation by $-2\frac{\kappa_{n-1/2}}{\Delta x}$, subtracting the second last equation and dividing by 2, we get

$$\frac{\kappa_{n-1/2}+\kappa_{n+1/2}}{2}u_n - \frac{\kappa_{n-1/2}+\kappa_{n+1/2}}{2}u_{n+1} - \frac{f_n}{2} = 0.$$

In this way, the Neumann version of the elliptic problem has the form:

$$\frac{1}{\Delta x^2}\begin{bmatrix} -\frac{\kappa_{-1/2}+\kappa_{1/2}}{2} & \frac{\kappa_{-1/2}+\kappa_{1/2}}{2} & & & \\ \kappa_{-1/2} & -(\kappa_{-1/2}+\kappa_{1/2}) & \kappa_{1/2} & & \\ & \ddots & \ddots & \ddots & \\ & & \kappa_{n-1/2} & -(\kappa_{n-1/2}+\kappa_{n+1/2}) & \kappa_{n+1/2} \\ & & & \frac{\kappa_{n-1/2}+\kappa_{n+1/2}}{2} & -\frac{\kappa_{n-1/2}+\kappa_{n+1/2}}{2} \end{bmatrix}\boldsymbol{u} + \begin{bmatrix} -f_0/2 \\ f_0 \\ \vdots \\ f_n \\ -f_n/2 \end{bmatrix} = \boldsymbol{0}$$

and consists in a set of $(n+3)$ non-linear equations, whose solution $\boldsymbol{u}$ can be approximated using the Newton-Raphson method (NRM). The initial guess used to start the iteration in NRM is obtained via a shooting method, thus reducing the boundary value problem given by the elliptic PDE in Eq. (11) to an initial value problem (IVP). A linear search is applied to find the shooting parameter, i.e. the initial condition of the IVP. Lastly, the solution of the IVP is approximated using the classical Euler's method for ODEs.

*Author contributions.* GDS conceptualized the paper, performed the numerical simulations and took the lead role in writing and revising the paper. JB, FF and TK conceptualized the paper and supervised the writing. All authors provided critical feedback and helped shape the research.

*Competing interests.* The contact author has declared that neither they nor their co-authors have any competing interests.

*Acknowledgements.* We are very grateful to the referee for reading carefully the paper and for their valuable comments. We acknowledge fruitful discussions with Valerio Lucarini and Robbin Bastiaansen. GDS would like to thank the Department of Mathematics and Statistics, University of Reading for its hospitality. TK and JB would like to thank the Scuola Normale Superiore for its hospitality. GDS is supoorted by the Italian national inter-university PhD course in Sustainable Development and Climate change. FF's research is funded by the European Union (ERC, NoisyFluid, No. 101053472). Views and opinions expressed are however those of the authors only and do not necessarily reflect those of the European Union or the European Research Council. Neither the European Union nor the granting authority can be held responsible for them.

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
