# Peer review of "Variational Techniques for a One-Dimensional Energy Balance Model"

_EGUsphere, 2023_

## Referee Comment (RC2)

Referee report

**Title:** Variational Techniques for a One-Dimensional Energy Balance Model

**Authors:** Gianmarco Del Sarto, Jochen Broecker, Franco Flandoli, and Tobias Kuna

**Journal:** Nonlinear Processes in Geophysics

General comments

The paper studies an energy balance climate model which consists of a nonlinear parabolic equation in one space dimension (on the interval $[-1, 1]$) with Neumann boundary conditions. This is a well-known model that describes the evolution of Earth's temperature based on the planet's energy budget. The impact of carbon dioxide on the system is represented by an additive parameter $q \in \mathbb{R}$, which is treated as a bifurcation parameter. The authors adopt a variational approach to prove the existence of steady-state solutions and relate the structure oif the minimizing set to the differentiability of the associated value function.

The topic of the paper fits well to the schemes of the journal. The results are convincing and original: relating the uniqueness of the solution of the stationary problem with the smoothness of the value function is a very nice idea. Moreover, the paper is clearly structured and makes good usage of images to illustrate the described approach.

Specific comments

1. Modelling the effect of the $CO_2$ on the energy budget as the additive constant $q$ is a choice that should be better motivated on physical grounds.

2. Removing the natural degeneracy of the diffusion coefficient at the boundary is more of a restriction than what the authors seem willing to admit on page 7. I understand that such a choice was made in order to reduce the complexity of the problem, but adapting the authors' approach to the real EBCM (degenerate parabolic equation) should be at least mentioned as an open problem.

3. In the context of optimal control, the value function is characterised as the solution of some nonlinear partial differential equation (the Hamilton-Jacobi equation). In this paper, it is shown that $-V'(q)$ equals the average of the minimizer of a certain functional. Could it be possible to characterize $V$ as the solution of some equation?

Technical corrections

This paper is essentially typo-free. I only found one typo:

1. just above line 90: enable*s*

---

## Author Comment (AC1)

**Reply on RC1**

January 21, 2024

We would like to thank the reviewer for his/her careful reading of the manuscript. We have taken almost all suggestions of the reviewer into account and hope that the manuscript is now acceptable.

In the following, there is a description of the changes made to improve the manuscript following the reviewer's suggestions.

**Major comment**

- " *Given the broad audience of the NPG journal, I think that the paper would be strengthened if the discussion about the implications of these results for climate science were presented in more detail [...] So, my recommendation is that the paper should be revised to strengthen the discussion of the physical interpretation of the results and their implications for climate science.*"

  We have completely rewritten the abstract to strengthen the discussion of the physical interpretation of the results. Moreover, we have added in lines 106-119 a more extended description of the physical interpretation of our results, recalling it also in the Conclusions section. Lastly, it is worth pointing out that in the updated version we have proved that the global mean temperature is non-decreasing with respect to the greenhouse gases concentration, i.e. the parameter $q$.

**Minor comments**

- "*L50 Could the authors provide more insight on the physical interpretation of parameter $\nu$ ?*"

  We have given a more thorough explanation of this, see added material in lines 52-55.

- "*L56 Using u as the zonally averaging temperature may be confusing. Why not use T\* or any other variant more consistent with the notation used in the 0D model?*"

  Thank you for the suggestion. The main equation of our work is the 1D-EBM and the associated elliptic problem. We decide to keep the notation

$u$ to denote the solution of PDEs, as done for instance in [1] and very common in math literature.

- "$u_{xx}$ is used in Equation 4, and $\Delta u$ is used later to represent the same quantity. Also, $u'$ is used instead of $u_x$ in this same equation."

  Thank you for pointing out these inconsistencies. We have uniformed the notation; $\Delta$ is reserved for the second partial derivative with respect to the space variable, $u'$ is reserved for the case when $u$ depends only on the space variable.

- " Eq. 5. Could the authors clarify the notation for the second term of the right hand side of this equation?"

  Thank you for the suggestion. Now the notation is explained in the line after that equation.

- "L70 It would be nice to point out that parameter $\kappa$ represents heat transport by the atmospheric dynamics, whose variability is known to be related to temperature gradients (an effect that is not explicitly accounted for in the simplified model)."

  Although the independence of $\kappa$ from $u$ (and its derivatives) is classical in EBMs literature, see for instance [1], we agree that it is a huge simplification with respect to the real dynamics. We have highlighted these facts in the lines next to the one suggested by the reviewer.

- " L125, where $q$ is assumed to be independent of latitude. It would be nice to include some discussion about how realistic this assumption is (e.g., https://agupubs.onlinelibrary.wiley.com/doi/full/ 10.1002/2017JD027221 )."

  In the new version (end of page 7) we have added the discussion about how nowadays it is not a state-of-the-art assumption, while it was the most common at the beginning of the century. The reasoning deals with the fact that $CO_2$ is considered a well-mixed greenhouse gas, i.e. it has a lifetime long enough to be relatively homogeneously mixed in the troposphere. Since we are considering a conceptual model, we think that it is reasonable to consider $q$ independent of latitude.

- "At the end of page 6, "is parameterized by a smooth, monotonically increasing function". Is this statement correct? Shouldn't the albedo be a monotonically decreasing function of the temperature?"

  Thank you for the correction. In the first draft of the paper, we have inverted the monotonicity property of the albedo.

- "L140 The covariance of $\mu$ is defined using $\Delta$ which is an operator. Could the authors clarify this aspect?"

  After Eq. (10), we have added the definition of the covariance operator for a Gaussian measure on a Hilbert space $H$. Then, given a covariance

operator, we have explained how to explicitly construct a random variable on $H$ such that it has a symmetric Gaussian law with that specific covariance operator.

- *"L147, could the authors clarify the meaning of the equation that is just before the sentence starting with "A rigorous..."?"*

The equation cited is the ODE:

$$y'(t) = R(y(t)). \tag{1}$$

Its fixed points are the values taken by the steady-state solutions of the space homogeneous 1D-EBM, which are constants. Since these values are more simply given by the roots of

$$R(y) = 0, \tag{2}$$

we have substituted (1) with (2).

- *"L156 Could the authors provide more detail on how these simulations have been performed to obtain the results shown in Figure 2?"*

To obtain the images in Figure 2, it is fundamental to approximate the steady-state solutions of the 1D-EBM. We used a finite difference scheme, which is described in Appendix A and leads to solving a set of non-linear algebraic equations. This system of equations is solved by the Newton-Raphson method (NRM). In the last three lines of the appendix, we have added how the initial condition for the iteration of the NRM is obtained.

**References**

[1] Michael Ghil. Climate stability for a sellers-type model. *Journal of Atmospheric Sciences*, 33(1):3 – 20, 1976.

---

## Author Comment (AC2)

**Reply on RC2**

January 21, 2024

We would like to thank the reviewer for the careful reading and the suggestions to improve the manuscript. We have taken them into account, as indicated in the answer below.

- "Question 1. *Modelling the effect of the $CO_2$ on the energy budget as the additive constant q is a choice that should be better motivated on physical grounds.*"

  We agree with the reviewer about the necessity to better explain the choice to model the radiative forcing $q$ of $CO_2$ as an additive term. We have motivated this choice by motivating it with a linearization argument involving the outgoing radiation $R_e$ depending on temperature $u$ and $CO_2$ concentration. The full argument can be found in Section 2.1 of the main manuscript.

- "Question 2. *Removing the natural degeneracy of the diffusion coefficient at the boundary is more of a restriction than what the authors seem willing to admit on page 7. I understand that such a choice was made in order to reduce the complexity of the problem, but adapting the authors' approach to the real EBCM (degenerate parabolic equation) should be at least mentioned as an open problem.*"

  The reviewer is right and we agree with him/her. The choice is made to be able to study with classical tools from calculus of variation the variational problem, from which all our results follow. We have clearly stated, in Section 2.1 (before the equation $\kappa(x) = D(1 - x^2) + \delta$, $D, \delta > 0$) and in Section 4, that it remains an open problem how to extend our results when $\kappa = \kappa(x)$ is degenerate at the boundary of the spatial domain.

- "Question 3. *In the context of optimal control, the value function is characterised as the solution of some nonlinear partial differential equation (the Hamilton-Jacobi equation). In this paper, it is shown that $-V'(q)$ equals the average of the minimizer of a certain functional. Could it be possible to characterize $V$ as the solution of some equation?*"

  We are not able to characterize the value function as the solution of an equation.

Indeed, continuing the parallel with optimal control, in that case the value function is, given an initial time and an initial state condition, the minimum value attained by the objective function. In our setting, the objective function corresponds to $F_q$. But $F_q$ does not depend on initial time and initial state condition.

We have preferred not to add any digression about the Hamilton-Jacobi equation to the manuscript, to avoid inserting a topic not investigated in the work.